# Design of a Methodological Intervention for Developing Respect, Inclusion and Equality in Physical Education

Antonio Muñoz-Llerena [1,2], María Núñez Pedrero [2], Gonzalo Flores-Aguilar [1,2,*] and Eloy López-Meneses [3]

1    Physical Education and Sport Department, University of Seville, 41013 Seville, Spain; amllerena@us.es
2    Research Group HUM-1061 "Inclusion and Innovation in Physical Education and Sport" (INEFYD),
     41013 Seville, Spain; maria.mabp@gmail.com
3    Department of Education and Social Psychology, Area of School Organization and Didactics,
     Pablo de Olavide University, 41013 Seville, Spain; elopmen@upo.es
*    Correspondence: gfaguilar@us.es

**Abstract:** The following educational intervention proposal arises from the importance of implementing an education based on fostering values through physical education (PE) lessons. PE has certain characteristics that contribute to enhancing learning at a social, affective and psychological level, in addition to promoting adequate physical development. The proposed design is based on Donald Hellison's personal and social responsibility model (TPSR), whose main objective is to achieve a teaching methodology that can convey values and skills in the lives of youth at risk of exclusion. Different sports modalities are used in the initiation phase, which make up a ten-week teaching unit and in which the game takes the leading role. The application of this program focuses specifically on students in compulsory secondary education, a stage in which significant changes are experienced in many aspects and levels. However, it is completely adaptable to other developmental stages. In this way, the main objective of this work is to create an intervention proposal that aims to promote, following a set of intervention units of sessions, the development of the three main values in which this work is based: respect, equality and inclusion.

**Keywords:** values development; TPSR; sport; adolescents

## 1. Introduction

Currently, sport is a great social phenomenon present in the lives of people in contemporary society that provides us with an important tool to influence, in a positive way, the lives of students. The promotion of values through the subject of physical education (PE) could play a very relevant role in different personal and future professional aspects of students.

Sport and physical activity are a magnificent vehicle for sensations, emotions and of those contents and values that society tries to transmit. Inculcating and promoting a life related to sport can bring numerous advantages and future virtues that will develop over the years [1]. Therefore, educational sport is an important tool for the integral formation of the individual and a way for the transmission of cultural values.

It is necessary to provide an education through PE that, on the one hand, helps students to know their own educational needs in a critical and reflective way, in addition to instilling principles and values beyond merely sports learning, while promoting their participation in teaching–learning processes from situations of observation, analysis or cooperation [2].

Research about values that arise from the practice of sports in young people shows that they are a fundamental element in society, helping and improving the quality of life of people in a physical, affective, social, cognitive and of a lifestyle way [1]. Therefore, it is essential to work so that these values have the place and importance they deserve in

the education of youth, in order to make them able to transfer the knowledge and skills acquired in the classroom to their daily lives.

### 1.1. Definition of Values

Before dealing with the development of values in PE lessons in school contexts, it is necessary to know, as a basis, the simplest concept of this study, which is the definition of "values".

Values are instruments that facilitate the construction of a social world, as well as the individual's interaction with the social reality in which he or she lives. Their development, from an early age and during the different stages of growth, allows for the construction of a more personal vision of reality and to be influenced by the surrounding environment [3]. They are also considered to be universal and enduring beliefs, in which a behavioral model or ultimate state of existence is personally and socially preferable to another behavioral model or opposite state of existence [4].

Madrigal [5] explains that values make it easier for us to define clear objectives in our lives, making possible the establishment of a mature and balanced relationship with the environment around us, becoming guides that lead to coherent behavior in different situations that may arise. He also states that values are complex and there are studied classifications, although their common purpose is to improve people's quality of life. Therefore, values are undoubtedly an excellent aspect to be employed in the educational environment.

The literature also emphasizes that "value" is associated with the person and his or her own existence, as it involves his or her conduct, structures and shapes his or her ideas and can also condition feelings. This is influenced by the process of socialization and what is internalized during this process, apart from the attitudes and ideas reproduced in the various socializing instances [6].

The literature affirms and assures that values arise through the emotional association of people, as they are linked to lived experiences and feelings, leaving aside intellectual logic. However, the intellect is not excluded, because sometimes the emotional image of a value is enriched after an intellectual effort [7].

### 1.2. Values and Physical Education

The presence of values in PE lessons can be a magnificent educational medium, as it brings together a series of special conditions focused on the most demanding objectives of pupils' education. A school of values is a suitable way for learning how to naturally behave in a social and affective way, contributing to psychological maturity and the harmonious development of the body [8].

Jiménez and Gómez [9] present in one of their studies the main factors that, in their opinion, have an impact on the development of values and attitudes at school. They distinguish two main contexts: the social context and the school context. In the social context there are three influential factors, which are family, media and society. The family is a key player as it is the first environment in which norms, attitudes and values are acquired. In this sense, Prat and Soler [10] argue that schools should be complementary and should in no way replace the role of the family. These three agents, in turn, encompass the school context, in which attitudes and values are mainly influenced by the teacher, the curriculum (including didactic aspects, methodology and assessment) and the class atmosphere.

The teacher's role and their methodology of teaching students is one of the most important factors, as their formation and awareness of the need to cover transversal contents through their lessons will be key for students to experience a real development of useful values in their daily lives. This is due to the fact that the relationship and the bond that are created, as well as the empathy, the tone of voice used, the communication, the looks, among other aspects, will be decisive in achieving the intended objectives and results [11].

PE is a unique subject that offers students constant interaction with each other. If the right tools are used and applied in the right way, depending on the subjects we work on

in the lessons, PE can become an excellent vehicle for transmitting values, attitudes and behavioral habits [12].

### 1.3. Respect, Equality and Inclusion

The formation of students and the educational system in which they are currently learning can entail a deep and unfinished debate. It is necessary for society to fight to build and promote an education of values that allows students to grow and improve from an optimistic, positive, motivating, non-judgmental and equal framework. With this, it is worth mentioning that PE offers great possibilities to explain and transmit what in a common classroom can go unnoticed, using sport to participate in the development of values, principles and attitudes necessary for the students' future lives: respect, equality and inclusion.

First, respect is an indispensable value in today's society that we can all encourage together. It is a value that allows us to learn to be considerate, tolerant and to consider the limitations, fears or insecurities of the people around us. Sport and PE can be an ideal context for students to respect, accept and learn behaviors such as assertiveness, collaboration or personal and social responsibility [13].

Second, equality implies breaking down the social barriers and structural obstacles that, today, still hinder men and women from having a place in society under conditions of real equality at different levels [14]. Its enhancement would be favored in an education free of sexist biases, in which people are treated equally and the perspective of what is conceived as "masculine" and "feminine" changes, so that no mention is made of positions of inferiority or superiority [15]. Generation after generation, numerous prejudices have been transmitted that have led to differences between boys and girls, reflecting an evident underdevelopment of supportive, tolerant and respectful sportive behavior [16]. Thus, fostering this value through PE can encourage this development to be transferred to society, seeking equality, which is undoubtedly everyone's goal.

Finally, if we refer to the concept of "inclusion" and therefore to an inclusive society, we are discussing the valorization of human diversity, the acceptance of people and the creation of real and fair opportunities for all [17]. It is necessary to differentiate this concept from that of "integration", which is understood as the incorporation into a group, i.e., to be part of a group, which in this case would be a group of students. It is related to concepts of "introducing", "involving" or "making part", which contribute to a group of people.

For the development of the aforementioned values, various interventions have been carried out in the literature with the objective of promoting them separately [18–23]. However, the work of Hellison [24] and his teaching personal and social responsibility model (TPSR) must be highlighted, which, through physical activity and sport, aims to bring about significant changes in the values of the participants. Pardo [25], in his doctoral thesis, provides more theoretical and practical information on how to work on the development of values through TPSR. His results show how there were significant changes in relation to factors such as respect, participation, effort or personal autonomy.

This research was based on Donald Hellison's TPSR [24], as well as on studies investigating its usefulness and its application with significant results, which can be worked on in PE classrooms with secondary school students [22,26–38].

### 1.4. Teaching Personal and Social Responsibility Model

In the 1970s, Donald Hellison, an academic at the University of Illinois (Chicago) and a well-known figure in PE and sports pedagogy, created the TPSR. It arose in the United States, and its main objective was to achieve a teaching methodology that could transmit values and life skills to socially excluded young people. It sought to encourage children to be involved with exercise through a framework of physical activity and both personal and social responsibility. TPSR has four key points: core values, levels of responsibility, program leader responsibilities and daily program format.

### 1.4.1. Core Values

In employing a program of this nature, it is necessary to describe the four core values that Hellison [24] outlines within the TPSR. The first of the core values is "putting kids first". It may have various interpretations, but its meaning is linked to helping children to be better people, thus promoting the second pillar, "human decency", which, together with the promotion of positive relationships, is necessary for the construction of a non-competitive and mutually supportive social framework for all. Thirdly, there is "holistic self-development". Hellison notes that physical development must take place alongside emotional, social and cognitive development in TPSR. Finally, the last core value is "a way of being", which seeks to convey that TPSR, more than a way of teaching, is a way of being, both for the teacher who leads the program and for the students, as they must be themselves at all times.

### 1.4.2. Levels of Responsibility

In TPSR, Hellison [24] distinguishes five levels of responsibility (Table 1), so that young people can take responsibility in PE programs and transfer acquired behaviors from the activity environment to other areas of life.

**Table 1.** Levels of responsibility.

| Levels of Responsibility | |
| --- | --- |
| Level I | Respecting the rights and feelings of others. |
| Level II | Effort and cooperation. |
| Level III | Self-Direction. |
| Level IV | Helping others and leadership. |
| Level V | Transfer outside the gym. |

At each level, the author proposes different strategies that enable the teacher and the student to achieve the objectives set out within each level. These strategic guidelines help, on the one hand, to increase and maintain the motivation of the students, together with the improvement of discipline and on the other hand, to ensure that the knowledge and attitudes worked on and developed are transferred to areas outside a gymnasium. In this work, the first level of responsibility is the one which will be mainly highlighted, as the aim is to promote aspects related to the value of respect in pupils and with strategies that allow, in turn, the complementary development of equality and inclusion.

In Level I, Hellison explains that respect can be defined as the core value of human decency, and its level is accompanied by three components: self-control, the right to peaceful conflict resolution and the right to be included and to have cooperative partners.

### 1.4.3. Program Leader Responsibilities

Having outlined the core values that make up this project, it is necessary to highlight the methodological pillars that support this intervention program, that is, the program leader responsibilities (Table 2). These should be integrated in each of the proposed sessions [24].

**Table 2.** Program leader responsibilities.

| Program Leader Responsibilities | |
|---|---|
| Gradual Empowerment | Adapt the degree of responsibility given to students in a progressive way. They must be given the opportunity to take responsibilities and learn to make good decisions. |
| Self-Reflection | Self-reflection is a necessary skill that accompanies empowerment. It is essential for development because all decision making and choices require reflection. |
| Embedding TPSR in the Physical Activities | In order for the program to be most effective, it is necessary that the teaching of physical and sporting activities is integrated with the teaching of social and personal responsibility. |
| Transfer | Students should be able to transfer the behaviors acquired in the PE lessons to other areas of life. |
| Being Relational with Kids | Creating a caring relationship with the students will be fundamental to the program. |

### 1.4.4. Daily Program Format

Strategies and methods for implementing the author's model are also presented, which seek to motivate students and keep them constantly interacting with the proposed program. Independently of the level we want to work on, Hellison [24] proposes a session structure with five components aimed at reinforcing each of the levels (Table 3). The functioning of the PE sessions should follow this structural pattern. In this way, learners are immersed in a routine process that will lead to faster progress in a work format they are familiar with and used to [39].

**Table 3.** Daily program format.

| Daily Session Structure | |
|---|---|
| Relational Time | The teacher–student relationship is established. The teacher interacts with them briefly, trying to demonstrate that he/she cares about them. |
| Awareness Talk | The purpose of this second part is to familiarize students with the objectives of the sessions and to inform them of the activities to be carried out during the practice. |
| Physical Activity Plan | This is the part of the session that takes up the most time. In it, the objectives proposed and integrated in the program are worked on through different physical sports activities. |
| Group Meeting | The members of the group meet again with the teacher to comment on what they have experienced that day. It is important to highlight positive aspects and share advice with the teacher based on how they have worked. |
| Reflection Time | This last section of the structure is dedicated to each student evaluating their behavior in the session and their responsibilities. |

Once having contemplated the importance of the development of values in the current educational system, the growing lack of values in society and the relevance of sport as a means of transmitting values [24,31,40–43], the main objective of this work is to design a methodological proposal easily adaptable to different sports and educational levels and development that seeks to jointly promote the values of respect, equality and inclusion through sport and physical activity in young people, within the formal training of compulsory secondary education (CSE), as well as promote their transfer to the daily lives of students. In this way, knowing those values and knowing how to use them in an

appropriate way will greatly benefit students in many social aspects and thus promote a lifestyle linked to sport and a healthy and balanced life.

## 2. Materials and Methods

### 2.1. Procedure

This didactic proposal has been designed through the use of different resources to provide the study with a solid scientific basis. First of all, a systematic bibliographic review was carried out to obtain scientific, good-quality literature about the different approaches existing in literature for working on respect, equality and inclusion, in order to know what programs, pedagogical models or ways of developing these values exist and to be able to choose one of them to design this program properly. On the basis of the existing literature found in the bibliographic review, TPSR was chosen. Subsequently, the context of implementation and the target population were analyzed.

Once the educational stage and the characteristics of the students that can generally be found in these educational contexts and target populations were known, a preliminary design was made, including the total duration of the intervention, the materials and spaces required, the possible sports modalities to be worked on, the structure of the sessions and the mechanisms of monitorization and evaluation of the implementation.

After the preliminary program was designed, the literature was reviewed again in order to ensure that the designed program concurred with the main characteristics of TPSR. Finally, an external expert with experience on TPSR in an educational context reviewed the final design.

Each of these phases will be explained below.

### 2.2. Systematic Review

The search was carried out within different well-known scientific databases in the field of sport and education: Web of Science, SCOPUS, SPORTDISCUS and ERIC, which were accessed via the institutional subscription of the University of Seville through the Spanish Foundation for Science and Technology (FECYT). The goal was to collect scientific articles and works related to values development in PE and the different ways, programs or pedagogical models utilized with each of the values in which this work focuses on (respect, equality and inclusion), in order to provide theoretical support based on empiric, valid results and methods to guide the present proposal.

In these databases, the following keywords were introduced: Sport; Value; Youth; Respect; Equality; Inclusion, crossing them with the Boolean operator AND. Different synonyms and Spanish translations were also utilized with the Boolean operator OR (Tables 4 and 5). Finally, only the studies directly related to the topic of the present work were selected.

**Table 4.** Keywords used in the literature search.

| Boolean Operator | AND | | | | | |
|---|---|---|---|---|---|---|
| | Joven | Deporte | Valor | Respeto | Igualdad | Inclusión |
| | Youth | Sport | Value | Respect | Equality | Social inclusion |
| | Adolescent | | | | Fairness | Social participation |
| OR | Young people | | | | Tolerance | Inclusive |
| | Teen | | | | Equal opportunity | Inclusión social |
| | Young adult | | | | Tolerancia | Participación social |
| | Adolescente | | | | Justicia | Inclusivo |
| | | | | | Igualdad de oportunidad | |

**Table 5.** Search queries employed in the literature search.

| Value | Search Query |
|---|---|
| Respect | respect OR respeto<br>AND<br>youth OR adolescent OR "young people" OR teen OR "young adult"<br>OR joven * OR adolescente<br>AND<br>sport * OR deport * |
| Equality | equality OR fairness OR tolerance OR "equal opportunit*" OR<br>igualdad OR tolerancia OR justicia OR "igualdad de oportunidad *"<br>AND<br>Youth OR adolescent OR "young people" OR teen OR "young adult"<br>OR joven * OR adolescente *<br>AND<br>sport * OR deport * |
| Inclusion | "social inclusion" OR "social participation" OR "social inclusive" OR<br>"inclusion social" OR "participación social" OR inclusive<br>AND<br>youth OR adolescent OR "young people" OR teen OR "young adults"<br>OR joven * OR adolescente *<br>AND<br>sport * OR deport * |

Adding * in a database search entails the use of truncation technique, which allows to broaden the search including different word endings and/or spellings.

### 2.3. Context Analysis

In the first instance, the intervention is designed to work in schools and would be carried out during PE classes. It is a methodological model designed to develop and promote moral and social values in the students through different sports modalities, working on them in initiation stages, constantly interacting with each other and reflecting on what they experience during the classes.

With regards to the target population of this project, the focus is on students in CSE, where PE classes are taught, generally, two hours a week. The age of the students will range from 11 to 16 years old, the period covered by CSE. During this stage of school (and life, in general), there are significant changes in many aspects of the person, physical, cognitive, affective, social, etc., that can affect the context that surrounds them in their daily lives and in which they develop [44].

One of the reasons why secondary school students should be the protagonists of this intervention program is because, in addition to being in a stage in which they experience difficult changes, there is a need to create useful and appropriate tools to awaken their motivation and carry out an action plan in which they feel they are the protagonists of their learning. It is important to empathize with them, understand them and make them feel part of "something", of a group in this case, respected and valued as each one of them is as a person, free and comfortable in the environment where they grow day by day, with their peers and with themselves.

The main objective of all sessions should be working on respect, inclusion and equal opportunities, as well as cooperative work among students to develop their knowledge and achieve common goals, regardless of the characteristics of each one.

Regarding the legislative framework that has been taken into account for the development of this proposal, it should be noted, first of all, that the current regulations grant the teaching staff certain flexibility to choose the methodology to be used during their educational interventions, as long as it attends to the nature of the competences indicated in the legal documents presented in the Official State Gazette (Organic Law 3/2020, of 29 December (LOMLOE) and Order ECD/65/2015) [45,46]. If we pay attention to the

educational regulations corresponding to the autonomous community of Andalusia, this is specified in a more adjusted form in the Order of 15 January 2021 [47].

Depending on the educational stage in which the proposal is to be carried out, all the curricular elements (objectives, contents, evaluation criteria, learning standards, key competencies, etc.) set forth in the aforementioned documentation must be adapted.

## 3. Results

This intervention proposal has been designed to work in schools and would be carried out during PE classes. It is a methodological model designed to develop and strengthen moral and social values in students through different sports modalities, working on them in initiation stages, constantly interacting with each other and reflecting on what they experience during the classes.

The model consists of 20 practical sessions of 50 min each one, designed with the aim of developing in the participants the aforementioned values: respect, equality and inclusion.

### 3.1. Design of the Intervention Proposal

It is necessary to know the educational context in which the methodological proposal is going to be developed, focusing the attention on CSE, an educational stage that completes the basic education of the students.

As explained in previous sections, an intervention proposal based on TPSR, which has the main objective of developing those values considered important by the authors (respect, equality and inclusion) in students, has been designed. The total duration of the intervention is 20 sessions, twice per week, with a length of 50 min each.

At the beginning of the program, before the sessions are carried out, there should be a previous explanation to the students in order to describe what it will consist of and what will be worked on during the time of the intervention. It is necessary to insist on the involvement and participation of the students, without going into much detail about the objectives of the program, so that students' behaviors can be analyzed under the naturalness and spontaneity of the participants.

After the brief explanation, each student will be given a questionnaire that he/she will have to fill in before starting, as well as at the end of the last session of the intervention. Specifically, the personal and social responsibility questionnaire (PSRQ) by Li et al. [48], in its Spanish version [49], will be used.

The methodological design of the intervention, as mentioned in previous sections, will consist of 20 sessions in which a total of six different sports modalities will be presented. The sports chosen in the proposal of the present work are athletics, volleyball, basketball, soccer, handball, intercrosse and a set of popular and traditional games. Most of them are well known sports and can usually be found in almost every PE syllabus.

A variety of modalities has been chosen that gives the possibility of using spaces usually found in schools. The materials to practice them are also affordable and easily found, which does not pose problems for carrying out the sessions. However, these sports modalities can be adapted and selected according to the needs of each school.

Finally, the evaluation and reflection on the overall functioning of the program has been selected. This will be carried out through tools such as diaries or questionnaires from the students, in addition to the collection of interviews and the recording of some of the sessions, which will collect information about what they have experienced throughout the intervention. On the other hand, field notes taken by the teacher during the sessions will also be useful.

Once the intervention period is over, there will be a large amount of information to evaluate and draw conclusions about the functioning of the proposal and its effectiveness.

*3.2. Development of the Didactic Proposal*

3.2.1. Objectives

The objectives of this didactic proposal aim to contribute to the development of values in PE lessons through sports practice. Those objectives are:

1. To develop and increase relational and social activities.
2. To make their best abilities available to the group.
3. To improve self-esteem, autonomy and self-confidence.
4. To work as a group, encouraging cooperation towards common goals.
5. To develop a sense of responsibility, collaboration and respect for others.
6. To respect and value others in the practice of these sports.
7. To value the team as a group and participation as a member.
8. Not to allow discrimination on grounds of sex or ability, and to value the recreational nature of the sport.
9. To respect the established rules and regulations, as a way of respecting both their teammates and the teacher.
10. To accept one's own and others' possibilities and limitations, as well as to develop self-esteem.
11. To relate and participate in group activities with other people, with attitudes of solidarity and tolerance.

3.2.2. Contents

The contents to be included in the present study are differentiated between three different types, depending on whether they are conceptual (representing knowledge, theoretical aspects or disciplines to be learned), procedural (representing the application to a practical situation of the conceptual material) or attitudinal (referring to the values, attitudes or norms generated by the practice):

- Conceptual.
  - Values education.
  - Habits of respectful, inclusive and equal behavior.
  - Respectful attitudes towards classmates and the teacher.
  - Cooperative work and achievement of common objectives.
  - Knowledge of the possibilities and limitations offered by one's body for the practice of physical and sporting activity.

- Procedural.
  - Adoption of non-discriminatory behavior towards other people.
  - Adoption of improvements in self-esteem, autonomy and self-confidence.
  - Interest in the practice of sport and its various advantages.
  - Development of values and expansion of social and relational activities.
  - Acceptance of one's own and others' limitations.
  - Execution of the technical gestures of these sports.

- Attitudinal.
  - Enjoyment of the motor possibility of one's own body through sport practice.
  - Appreciation of the group as a team and their participation as a member.
  - Appreciation of the personal relationships created in the educational sports environment.
  - Appreciation of the space and the material used.

3.2.3. Evaluation Criteria

The evaluation of the didactic proposal will be carried out based on the students' implication. It is important to note that the specific evaluation criteria will be chosen depending on the academic year in which this proposal is implemented, so the legal documents mentioned in Section 2.2 will have to be considered.

In this project, similar to Pardo [25], a personalized document is used for each partici-
pant that requires dedication and special attention throughout the intervention: the diaries.
The diaries will be documents drawn up by the students. For each session carried out,
they should describe the ideas, reflections, emotions and thoughts that arose from their
experience in the practice of the different sports modalities. In addition, a self-evaluation
section will be added, which addresses behavior and participation during the session.

### 3.3. Timeframe of the Intervention

The present intervention is designed for a specific period of time. There will be up to
20 sessions (Table 6), which are divided into six different sports modalities, as well as a
final section of popular and traditional games.

**Table 6.** Intervention didactic unit.

| Session Number | Content |
| --- | --- |
| 1 | Initiation athletics i |
| 2 | Initiation athletics ii |
| 3 | Initiation athletics iii |
| 4 | Initiation voleyball i |
| 5 | Initiation voleyball ii |
| 6 | Initiation voleyball iii |
| 7 | Initiation basketball i |
| 8 | Initiation basketball ii |
| 9 | Initiation basketball iii |
| 10 | Initiation football i |
| 11 | Initiation football ii |
| 12 | Initiation football iii |
| 13 | Initiation handball i |
| 14 | Initiation handball ii |
| 15 | Initiation handball iii |
| 16 | Initiation intercrosse i |
| 17 | Initiation intercrosse ii |
| 18 | Initiation intercrosse iii |
| 19 | Popular/traditional games I |
| 20 | Popular/traditional games II |

### 3.4. Methodological Strategies

This didactic proposal is oriented to the development of values in the school context,
focusing on students in CSE. It is crucial to pursue the active participation of students,
promoting motivation and enjoyment of the sports experienced.

In each session, all the contents stated in the syllabus are developed, taking into
account the legislative framework previously explained. The structure and functioning of
the sessions derive from the structure of the TPSR session: 1. relational time; 2. awareness
talk; 3. physical activity plan; 4. group meeting; 5. reflections. All sessions follow the
core values and program leader responsibilities of this model, previously explained in
Section 1.4. In addition, the specific methodological strategies proposed by Hellison for the
different levels of responsibility are used. In the case of the present work, which focuses
mainly on level 1 of responsibility, the methodological strategies employed are centered
on modifying the rules of the games (adapting the games so that everyone can participate
on equal terms) or on the elaboration of groups or teams within the tasks of the session
(students have the rule of making the sides fair).

It is necessary to reiterate that each sport is worked in initiation stages, so that the
activities are exposed, mostly, in a playful way, using recreational and playful exercises for
the promotion of cooperative work and thus of the relationships between students.

Three sessions of each sport should be carried out, with a total length of 50 min, the
standard duration of PE classes in the Spanish educational context, as well as the one
used by Pardo [25] in his interventions. In addition, it is important to note that all sport

modalities in initiation entail a (third) session especially dedicated to adapted and inclusive sport. To facilitate the understanding of these concepts, it is necessary to clarify that there are differences between them. Adapted sport is a tool that allows people with disabilities to participate safely in a particular sport [50]. Inclusive sport, on the other hand, is the practice that enables students with and without disabilities to participate together in a sport [51]. Therefore, in the third session of each modality, there will be several exercises adapted to the specific sport, which aim to make students experience different sensations and empathize with those who have different abilities from them.

An important factor to consider when carrying out an intervention program of these characteristics in PE classes are the conflictive situations and problematic circumstances that could arise, such as exclusion or inequality among them. Because of that, one of the methodological strategies that should be employed throughout the intervention is conflict resolution, which is defined by Ruiz Omeñaca [52] as a cognitive–affective–behavioral process by which a person, or a group, finds or identifies the most appropriate means to deal with the problems that life may present. Hellison [24], for his part, shares in his work a chapter dedicated to the description of different strategies, divided among the levels in which he works, with the aim of solving specific problems and situations in practice. If we focus on the first level of responsibility (respect), which is the main focus of this intervention, there are a number of possible strategies that can be used to remedy lack of participation, exclusion among peers, equality or conflict resolution [24,25]. The different strategies proposed in this work are shown below (Tables 7 and 8).

**Table 7.** Strategies for solving conflicts related to individual behaviors.

| Strategy | What Does It Consist of? |
| --- | --- |
| Accordion Principle | Depending on the behavior of the students, the duration of an activity they like to take part in is reduced or extended. It is necessary to try to individualize, as it would be unfair for the whole group to be punished for the bad behavior of others. |
| Logical Consequences | Depending on the problem, the disruptive student may apologize publicly to his or her classmates or his or her behavior may be monitored, and a fair solution may be reached that is acceptable to both the student and the person concerned. |
| Negotiation | This strategy tries to reach an agreement during the conflict. |
| Sit-Out Progression | It is aimed at those who are more confrontational. They are given the option of not participating in the lesson and rejoining the class later. If this is not effective, the teacher should try to negotiate with the student to solve the problem. Finally, failing that, the student should be referred to a specialist. |
| No Plan, No Play | It is similar to the group meeting but takes place during the execution of the activities. If there is a problem or conflict in the lesson, the teacher brings the students together and they try to find a solution. |
| Grandma's Law | The students are offered to play or to practice the activity they like the most, on the condition that they are not allowed to complain. In this case, the activity stops, and they move on to another activity proposed beforehand (which may not be to their preference). |
| Teacher-Directed Group | It is based on creating two different work groups, differentiating the components by whether they are conflictive or have a great capacity for personal autonomy. The conflictive group will work with the teacher and the other will work independently. It is possible to change groups if they show responsible and autonomous behavior. |
| Five Clean Days | The aim is to assess whether a student makes progress and is able to show respect and participation for five or more consecutive days. |
| Referral | Offer the student the service of specialists in difficult situations and with the ability to deal with subjects with special needs. |

**Table 8.** Strategies for solving collective conflicts.

| Strategy | What Does It Consist Of? |
|---|---|
| Sport Court | The group chooses three students to make decisions in case of disputes where the group cannot reach an agreement. |
| Self-Officiating | The aim is to encourage students to be able to take responsibility and solve conflicts by themselves without a mediator as a referee. |
| Talking Bench | Locate a specific place in the lesson so that there is a dialogue between two students to try to solve problems. |
| Emergency Plan | This is based on developing a contingency plan in case a conflict arises during the activity (e.g., coin game: heads or tails). |
| Making New Rules | Students are responsible for preparing a list of rules for the whole class. These need to be redone in case they are not effective. |

In this proposal, what is intended is that students are able to resolve conflicts and adverse situations, allowing them to be in charge of the conflict resolution process autonomously, reaching common and fair agreements for all, always focusing on attitudes derived from respect, equality and inclusion towards all classmates. Hence, sport will become an essential tool that will allow for the creation of varied situations, always seeking to increase and build attitudes and behaviors appropriate to each of them. The games are also an ideal context for many of the learning processes to take place and are therefore the didactic resource par excellence. The games focused on sports initiation that students will experience throughout the intervention will enable them to experience powerful changes and improvements related to the reduction of their less desirable behaviors and the acquisition of positive habits and behaviors.

*3.5. Description of the Sessions*

In this section, the functioning of the contents of the three different sessions of the same sport are explained. Only one sport discipline is shown in this section (athletics) (Tables 9–11), due to space issues. The other disciplines (the remaining 17 sessions) are utterly explained in the Supplementary Materials Table S1–S17.

**Table 9.** Session I, Athletics I.

| Sport:<br>ATHLETICS I | Duration:<br>50 min | Materials:<br>16 Large Hoops, 15–20 Handkerchiefs,<br>24 Large Cones. | Participants:<br>24–30 |
|---|---|---|---|
| **Objectives:**<br>To make contact with the modality of athletics.<br>To perform exercises to assimilate the basic running technique.<br>To use the game as a tool for learning a technique.<br>To develop basic motor skills through relay races. | **Objectives related to the program:**<br>To create balanced teams during the activities.<br>To respect each other, the rules, the material and the space.<br>To increase the participation of all students. | **Strategies for motivation:**<br>Give them a choice between the warm-up game or running around the court.<br>Ask if anyone wants to be responsible for the material during the session. If not, name someone at random. The chosen one can select a partner to help him/her. | |

| **Relational time and awareness talk (4–5 min)** |
|---|
| The teacher gathers the students in a circle seated on the floor and welcomes them, asks how they are, asks about the mood of the students and may share a personal anecdote or something that has recently happened to him/her and which he/she would like to tell them. Then, he/she gives an overview of the program in which they are going to participate. Then, he/she explains the content of the session and the objectives. Before starting with the activities in this first session, the teacher will ask the students to create a list of "rules for the PE lessons". Each student will have to write one and, the next day, there will be a sharing session to select and clarify the rules to be respected during the classes. Insist from the very beginning on punctuality for the sessions, the importance of involvement and appropriate clothing.<br>***Strategies used:*** *awareness raising → students are informed of the objectives of the program, which are related to the main objectives (respect, equality and inclusion); establishment of rules → a set of rules is established to be taken into account during the session; material manager → one or more pupils are designated to be in charge of the material.* |

**Table 9.** *Cont.*

| Physical Activity Plan—Warm-up (8 min) | | |
|---|---|---|
| **Description** | **Key Aspects** | **To Be Taken into Account** |
| Activation: 2 running laps around the court. | | |
| Thread cutter: One partner goes after the others. The game consists of the player chasing a teammate and cannot go after another teammate unless he crosses in between them. **Variant:** Increase the number of players chasing their teammates. | Dynamism, constant movement. Differentiate well the partner who is left behind so as not to cause confusion. | *It is important that all students participate in the activity and are integrated in the lesson. By including more chasers in the activity, participation rises.* **Strategies used:** *modification of the rules to enhance participation and keep the group motivated.* |
| Articular mobility with displacement | - | *In this first session, the warm-up will be carried out by the teacher. However, it would be important to tell the students that, throughout the battery of sessions, they will have to take care of it by themselves. Each day a different student will be in charge.* |
| **Physical Activity Plan—Principal Part (30 min)** | | |
| Activity 1: Baby chicken (5 min) Groups of 5. One is the chicken in the middle. The other 4 are placed in 4 corners. They must pass through the center, close to the chicken, performing a running technique exercise proposed by the teacher without making any noise. Techniques: <br>• Skipping in front one leg. <br>• Skipping in front with two legs. <br>• Skipping behind one leg. <br>• Skipping behind two legs. | If the chicken touches someone, there is a change of role. Maximum speed to change corners. Correct technical gesture. Corrections in case of mistakes (feedback). | *The teams in this type of activity should be randomized, so that students do not always share a team with the teammate with the closest affinity. It would be advisable for the teacher to spontaneously create the groups. This can be carried out, for example, by giving a number (in this case, from 1 to 5) to each student, and then they would be grouped according to the number assigned to them.* |
| Activity 2: The 4 corners (8 min) Same distribution as the previous exercise. A hoop is placed in each corner and a player inside it. At teacher's voice, change hoop or corner by performing the corresponding technique. The teammate in the center must move into one of the free corners to become free. | Dynamism, constant movement. | - |
| Activity 3: Hunters and preys (8 min) Small space of 20m × 20m. The prey team wears a handkerchief at the back of their trousers. The hunters will have to catch the handkerchiefs. **Variant 1:** Modify the playing space. **Variant 2:** Count the time each team spends to complete the activity. Whoever obtains all the handkerchiefs first wins. | Dynamism, constant movement. | *There will be 2 teams, and they will be formed by two captains (boy/girl). The team choice will be varied, and they will take turns choosing, so that the turn is respected, and the boy/girl will be chosen according to their turn.* |

**Table 9.** *Cont.*

| | | |
|---|---|---|
| Final activity: Relay race (8–9 min)<br>Teams of 4 stand at one end of the field. There are four rows of cones on the court. The first teammate moves to the first cone and back, the second to the second cone and back and so on until the last one is reached. The witness exchange zone (which will be a handkerchief in this case) will be when they reach the starting point again. The team that completes the whole course wins.<br>**Variant:** Change the distances of the game. | The next partner does not start until the previous one touches his hand. | *Competitiveness between teams must be controlled. Encourage cooperative work and do not allow disrespect towards colleagues. In the case of great inequalities between groups, give students the opportunity to change groups, always in dialogue and reaching a consensus between members.*<br>*Strategy used: autonomy: students must learn to work on their own and to solve adverse situations.* |

**Group meeting and reflection time (8 min)**

The teacher gathers the students in a circle, sitting on the floor as at the beginning of the session. He/she thanks them for their participation and proceeds to reflect on what happened during the sessions, trying to evaluate the behavior of the participants and how the session was carried out. For the reflection, there can be questions such as: Which activity did you like the most? How do you think the session went? Would you have changed anything in the session? What do you think your participation has been like today? What did you learn today? What do you think could be useful for you in your daily life? How could you put it into practice? Finally, a blank sheet of paper is handed out on which the students write positive and negative things about what happened. In addition, they self-evaluate themselves using a color-coded system (red for bad participation/behavior, yellow for average participation/behavior, green for good participation/behavior or blue for excellent participation/behavior). They should highlight if something was significant for them and can add constructive criticism for improvement in the next class. They will keep this sheet of paper and write a new one each session. Finally, the students in charge of the material take care of their duties.
*Strategies used: In reflection time, they have to describe their behavior during the practice. In this way, they are involved in the evaluation process.*

These images are graphical representations of the setup/put into practice of the activities, which aim to help readers understand how the activities are organized and carried out. They have been made by authors using Paint software and Microsoft Powerpoint software.

**Table 10.** Session II, Athletics II.

| Sport:<br>ATHLETICS II | Duration:<br>50 min | Materials:<br>35 Hoops, 10 Training Vests (Same Color), 4 Batons, 9 Large Cones, 8 Small Cones and 3 Sticks. | Participants:<br>24–30 |
|---|---|---|---|
| **Objectives**:<br>To familiarize with the modality of athletics, specifically with relays.<br>To perform exercises to assimilate the basic technique of the relay race.<br>To use the game as a tool for learning a technique.<br>To develop basic motor skills through relay races. | **Objectives related to the program**:<br>To create balanced teams during the activities.<br>To respect each other, the rules, the material and the space.<br>To increase the participation of all students. | **Strategies for motivation**:<br>Give them a choice between the warm-up game or running around the court.<br>Ask for a person responsible for the material today, otherwise choose someone randomly. He/she can select a partner to help him/her.<br>Ask if there are any volunteers who want to lead the warm-up. As it is the second session, it will be voluntary, but from this session onwards, it will be one student per day. | |

**Relational time and awareness talk (4–5 min)**

The teacher gathers the students in a circle seated on the floor and welcomes them, asks how they are, asks about the mood of the students and may share a personal anecdote or something that has recently happened to him/her and which he/she would like to tell them. The teacher also congratulates the students if they have been punctual and are dressed correctly; if not, he/she corrects them. Next, the rules are shared in order to create a list of "rules for the PE class". Once the rules have been established, the content of the session is presented, so that the objectives of the session are explained.
*Strategies used: awareness raising → students are informed of the objectives of the program, which are related to the main objectives (respect, equality and inclusion); establishment of rules → a set of rules are established to be taken into account during the session; material manager → one or more pupils are designated to be in charge of the material.*

**Table 10.** *Cont.*

| Physical Activity Plan—Warm-Up (8 min) | | |
|---|---|---|
| **Description** | **Key Aspects** | **To be Taken into Account** |
| <u>Activation:</u> 2 running laps around the court. | - | - |
| <u>Pac-man</u>: Free distribution, no teams. Use of batons to catch teammates, who move only and exclusively along the lines painted on the court. The baton is given to the player caught from behind (simulating the handing over of the baton on the court in real sport). **Variant**: One more baton is introduced into the game. | It is important that the player who has the baton hands it over, simulating a real relay handover. The rest of the players will be activated, as this is a very dynamic activity. Respect the route of the lines. Do not go out. | *It is important that all students participate in the activity and are integrated in the lesson. By including more batons in the activity, participation increases and students have more stimuli to attend to on the court.* ***Strategy used:****modification of the rules to enhance participation and keep the group motivated.* |
| <u>Articular mobility with displacement</u> | - | *If a student is a volunteer to lead the warm-up, congratulate and thank him/her for his/her involvement in the session. The teacher should guide and give guidelines to the student in this responsibility.* ***Strategy used****: autonomy: give learners responsibilities and tasks that they have to carry out by themselves.* |
| **Physical Activity Plan—Principal Part (30 min)** | | |
| Activity 1: Rock, paper, scissors (5 min) Two teams are formed. Set up 35 hoops in a row. Each team will have a baton and will place it at one end of the row of hoops. On a signal, the first in the line runs out until they face their opponents. It is a rock, paper, scissors duel. Whoever wins goes on, and whoever loses goes back. The aim is to move the baton to the other end of the line of hoops. The starting movements can be varied (limping, skipping, jumping, etc.). | The second in the row must be ready to go out and pick up the baton in case his/her partner loses. | *Same dynamics to form teams: They will be formed by two captains (boy/girl). The choice of the team will be varied, and they will take turns choosing so that the turn is respected, and they choose boy/girl according to their turn.* |
| Activity 2: I pass the baton to you (10 min) Placed in rows of 8 students, the baton will be passed using an ascending handover technique to the last person in the line, who will run to the first person in the line. The objective is to reach a previously set goal line. **Variant:** Using descending handover technique. In this technique, the baton is handed over by making a descending movement of the hand to hand over the baton. | Do not place too far apart. Sufficient space for baton passes. The next one will only start if the partner arrives at the starting point. Good baton handover is important. | - |

**Table 10.** *Cont.*

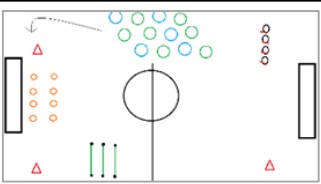

Final activity: Circuit (15 min)
Teams of 4, standing at one end of the court. Each participant goes through the circuit around the court (jumping inside hoops, performing a zigzag through the cones, jumping stick-made hurdles and sprinting), and when they reach again the starting point, there will already be another teammate ready to start, and they will have to use an ascending handover technique to make the relay. The baton exchange area will be on the beginning, at the chosen end of the court. The team that completes the whole race before the other wins.
**Variant**: Use a descending handover technique.

Each player must pass behind the cones placed on the court. Avoid unevenness in the race. The next players to start must be ready and they will only start when they have the baton.

*Competitiveness between teams must be controlled. Encourage cooperative work and do not allow disrespect towards colleagues. In the case of great inequalities between groups, give students the opportunity to change groups, always in dialogue and reaching a consensus between members.*
***Strategy used****: autonomy: students must learn to work on their own and to solve adverse situations.*

**Group meeting and reflection time (8 min)**

The teacher gathers the students in a circle, sitting on the floor as at the beginning of the session. He/she thanks them for their participation and proceeds to reflect on what happened during the sessions, trying to evaluate the behavior of the participants and how the session was carried out. For the reflection, there can be questions such as: Which activity did you like the most? How do you think the session went? Would you have changed anything in the session? What do you think your participation has been like today? What did you learn today? What do you think could be useful for you in your daily life? How could you put it into practice? Finally, a blank sheet of paper is handed out, on which the students write positive and negative things about what happened. In addition, they self-evaluate themselves using a color-coded system (red for bad participation/behavior, yellow for average participation/behavior, green for good participation/behavior or blue for excellent participation/behavior). They should highlight if something was significant for them and can add constructive criticism for improvement in the next class. They will keep this sheet of paper and write a new one each session. Finally, the students in charge of the material take care of their duties.
***Strategies used:*** *In reflection time, they have to describe their behavior during the practice. In this way, they are involved in the evaluation process.*

These images are graphical representations of the setup/put into practice of the activities, which aim to help readers understand how the activities are organized and carried out. They have been made by authors using Paint software and Microsoft Powerpoint software.

**Table 11.** Session III, Athletics III.

| Sport: ATHLETICS III | Duration: 50 min | Materials: 20 Bandages. 20 Handkerchiefs. 2 Large Cones. 32 Small Cones. 10 Hoops. 2 Steps. 2 Hurdles. | Participants: 24–30 |
|---|---|---|---|
| **Objectives**: To make contact with the modality of athletics. To use the game as a tool to promote cooperative work and inclusion. | | **Objectives related to the program**: To create balanced teams during the activities. To respect each other, the rules, the material and the space. To increase the participation of all students. To integrate all classmates in the lesson. To avoid offensive verbal and physical behavior. | **Strategies for motivation**: Ask if anyone wants to be responsible for the material during the session. If not, name someone randomly. The chosen one can select a partner to help him/her. Ask if there are any volunteers who would like to lead the warm-up. If not, name someone. |

**Table 11.** *Cont.*

| Relational Time and Awareness Talk (4–5 Min) | | |
|---|---|---|
| The teacher gathers the students in a circle seated on the floor and welcomes them, asks how they are, asks about the mood of the students and may share a personal anecdote or something that has recently happened to him/her and which he/she would like to tell them. Then, he/she gives an overview of the session and the objectives, emphasizing the importance of helping classmates. Before starting with the activities in this session, the teacher will ask the students to create a list of "rules for the PE lessons". Each student will have to write one and, the next day, there will be a sharing session to select and clarify the rules to be respected during the classes. Insist from the very beginning on punctuality for the sessions, the importance of involvement and appropriate clothing. *Strategies used: awareness raising → students are informed of the objectives of the program, which are related to the main objectives (respect, equality and inclusion); establishment of rules → a set of rules are established to be taken into account during the session; material manager → one or more pupils are in charge of the material.* | | |

| Physical Activity Plan—Warm-Up (8 min) | | |
|---|---|---|
| **Description** | **Key Aspects** | **To Be Taken into Account** |
| <u>Activation:</u> 2 running laps around the court. <u>Articular mobility with displacement</u> | | *Congratulate and highlight the important role of the student leading the warm-up. The teacher must help and guide the student in this task. Do not allow disrespect.* |

| Physical Activity Plan—Principal Part (30 min) | | |
|---|---|---|
| Activity 1: My other half (5 min) In pairs, the students must agree on a keyword between them. One puts on the blindfold and the other prepares to stand somewhere in the playground. On the teacher's signal, the student with the blindfold goes out to look for his or her half, who will be shouting the agreed word to find him or her. Once the blindfolded one finds the partner, they change roles. **Variant:** Competitive. Whoever finds their half first wins the game. | The partner without the blindfold is not allowed to move after the teacher's signal. | *Initially, the pairs can be formed freely by the students. Throughout the game, at the teacher's signal, they will have to change partners. It is not possible to repeat with the same partner.* ***Strategy used:*** *modification of the rules to enhance participation, inclusion and keep the group motivated.* |
| Activity 2: Busted (5 min) We continue in pairs and with the blindfolds. This time, two pairs keep the blindfold, while the others spread out in the middle of the court. The teammates without a blindfold guide the blind ones, who will try to catch the others. Those who are not blindfolded will try not to be caught. If they are caught, they change roles. | A handkerchief shall be used to differentiate the pursuers. | *It is important that all students participate; therefore, an extra pair will be included to increase participation in the activity.* ***Strategy used:*** *modification of the rules to enhance participation, inclusion and keep the group motivated.* |
| Activity 3: I trust you (8 min) Using the width of half-court, two teams are formed. Two large cones are placed in the middle with a handkerchief (or ball) on top, one for each team. The aim of the game is similar to the "handkerchief": The teacher says a number, and the player of each team with this assigned number will try to catch the handkerchief before the other team. They will be blindfolded, and the teammates have to guide them. | Just take the handkerchief. No need to return to the starting line. The first one to succeed wins a point. | - |
| Final activity: Blindfolded circuit (12 min) There are two circuits formed with diverse materials, and the students will have to complete them blindfolded (one circuit per team). It consists of a zigzag in cones, hurdle crossing (passing underneath), jumping a step with two feet, searching for a cone to place in the hoop and the final sprint along the court. In pairs, the blindfolded one is guided by the partner, and they can touch each other. **Variant 1**: The partner goes to the blindfolded one's side, but they cannot touch each other. **Variant 2**: Relay competition with a pair from the opposite team. | Students must be careful with the indications. | 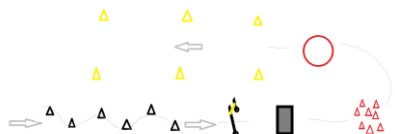 *Competitiveness between teams must be controlled. Encourage cooperative work and do not allow disrespect towards colleagues. In this game, which includes a little more risk, insist on being careful in the indications given to teammates.* ***Strategy used***: *autonomy: students must learn to work on their own and to solve adverse situations.* |

**Table 11.** *Cont.*

| Group Meeting and Reflection Time (8 min) |
| --- |
| The teacher gathers the students in a circle, sitting on the floor as at the beginning of the session. He/she thanks them for their participation and proceeds to reflect on what happened during the sessions, trying to evaluate the behavior of the participants and the session itself. For the reflection, there can be questions such as: Which activity did you like the most? How do you think the session went? Would you have changed anything in the session? What do you think your participation has been like today? What did you learn today? What do you think could be useful for you in your daily life? How could you put it into practice? Finally, a sheet of paper is handed out on which the students write positive and negative things about what happened. In addition, they self-evaluate themselves using the color-coded system. They should highlight if something was significant for them and can add constructive criticism for improvement in the next class. They will keep this sheet of paper and write a new one each session. Finally, the students in charge of the material take care of their duties. *Strategies used: In reflection time, they have to describe their behavior during the practice. In this way, they are involved in the evaluation process.* |

These images are graphical representations of the setup/put into practice of the activities, which aim to help readers understand how the activities are organized and carried out. They have been made by authors using Paint software and Microsoft Powerpoint software.

## 4. Discussion and Conclusions

This proposal is the result of the concerns that arise when questioning how important PE is in schools and the many tools it offers us, learning and appreciating how it can contribute to the transmission and development of those essential values of life in students, as they contribute to the integral construction of a person. This work has highlighted the importance of PE in schools and its great capacity to integrate a teaching based on values. A didactic proposal has been designed, which can provide a vast array of benefits to the participants through sports activities. The presence of PE in the school curriculum is necessary, both to motivate students to include sport in their lives and to promote their healthy lifestyle. During PE classes, students are in constant contact with each other, and this situation can be an optimal way to interiorize those positive values, making them able to transfer what they learn to everyday situations and different social contexts, so that they grow up and become responsible, manage their own emotions and feelings and maintain appropriate attitudes in all areas of their lives.

It is a common belief that sport contributes to the acquisition and learning of values and life skills that will make participants able to face their future lives with security [53]. However, participation alone in a sport activity does not mean that those values will be learned [54]. Those benefits might arise when the social context (family, peers and teachers/coaches) contributes to the sportive experience in a proper way [41–43,55,56]. Following Theokas et al. [57], both sports and life skills do not rise up alone, but a certain framework must be followed in order to learn, enhance and transfer sports and life skills. This particular framework, according to the literature, should include a proper program climate together with specific contents [31] and a mastery-oriented and caring climate, as well as supportive relationships and opportunities to learn and put into practice values and life skills [43]. In this way, this didactic proposal provides a well-structured framework that meets the previously stated criteria, designed based on a proven and successful pedagogical model such as TPSR [24].

Usually, these programs are not explained in detail or well expounded in the literature, and sometimes it can be difficult for teachers to put them into practice, especially when they do not have much time to research and go in depth into the core of those programs. This proposal has been designed in order to help teachers to easily adapt TPSR to their educational level, with a set of different sports and material that can be utilized in every school and with clear methodological instructions, which will contribute to the development of values in their students.

It should be noted that the responsibility for the functioning and good development of classrooms does not fall mainly and exclusively on the students. The role of the teacher requires the same level of involvement in this type of project [24]. It is essential for the teachers to present vocation, enthusiasm and, above all, the interest and desire to achieve

a significant change in the learning and the lives of their students and, in general, in the current education system and society.

Nowadays, a series of undesired behaviors and habits such as competitiveness, consumption, individualism, greed or envy, among others, prevail. Consequently, the world is impacted by the striking "crisis of values" that builds an increasingly less human society with a great lack of motivation, sensitivity, ideals and a lack of critical education in terms of social values [58]. The achievement of the goals of the present proposal could mean a radical change in the lives of students, as they would be able to acquire the capacity to face a world full of complexes and prejudices that society promotes every day.

In this work, guidelines and parameters from the literature have been followed, taking into account previously made research and interventions in order to design the present proposal [19,22–25,34,59]. Furthermore, an expert with more than five years of expertise in TPSR reviewed the framework of the proposal. In this sense, the authors consider it to be well-founded. However, it has not been carried out, and implementation is a crucial factor on the effectiveness of an intervention [60].

The aim of this work was to design a didactic proposal to help to promote the development of the values of respect, equality and inclusion in CSE through PE, making students reflect on their actions inside and outside the classroom in order to foster transference of those values to other social contexts. The authors believe that this proposal will be helpful for teachers to enhance their teaching in terms of values development, contributing to a better society as long as the methodological framework is followed. Giving students a voice and a leading role in their own learning process, as well as helping them to recognize those behaviors based on respectful attitudes that do not show gender discrimination or discrimination against people with disabilities, will be a useful starting point to promote constructive opinions in pursuit of interiorizing this positive behavior so that it becomes part of students' daily lives, making them better persons and helping to build a better society.

**Supplementary Materials:** The following are available online at https://www.mdpi.com/article/10.3390/su14010390/s1, Table S1: Session 4, Volleyball I; Table S2: Session 5, Volleyball II; Table S3: Session 6, Volleyball III; Table S4: Session 7, Soccer I; Table S5: Session 8, Soccer II; Table S6: Session 9, Soccer III; Table S7: Session 10, Handball I; Table S8: Session 11, Handball II; Table S9: Session 12, Handball III; Table S10: Session 13, Basketball I; Table S11: Session 14, Basketball II; Table S12: Session 15, Basketball III; Table S13: Session 16, Intercrosse I; Table S14: Session 17, Intercrosse II; Table S15: Session 18, Intercrosse III; Table S16: Session 19, Popular and traditional games I; and Table S17: Session 20, Popular and traditional games II.

**Author Contributions:** Conceptualization, A.M.-L. and M.N.P.; methodology, A.M.-L. and M.N.P.; validation, A.M.-L., G.F.-A. and E.L.-M.; formal analysis, A.M.-L.; investigation, A.M.-L. and M.N.P.; resources, G.F.-A.; data curation, G.F.-A.; writing—original draft preparation, M.N.P. and G.F.-A.; writing—review and editing, A.M.-L and E.L.-M.; visualization, E.L.-M.; supervision, E.L.-M.; project administration, A.M.-L.; funding acquisition, A.M.-L. All authors have read and agreed to the published version of the manuscript.

**Funding:** This research has been partially funded by the Innovation and Inclusion in Physical Activity and Sports (INEFYD) research group (HUM-1061).

**Institutional Review Board Statement:** Not applicable.

**Informed Consent Statement:** This work is a didactic proposal that has not been put into practice, thus no informed consent has been required.

**Data Availability Statement:** Not applicable.

**Conflicts of Interest:** The authors declare no conflict of interest.

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
