# Peer review of "Design of a Methodological Intervention for Developing Respect, Inclusion and Equality in Physical Education"

_sustainability, doi:10.3390/su14010390_

Round 1

Reviewer 1 Report

General Comments

First, I would like to recognize the authors for developing this educational intervention proposal, which focuses on promoting respect, inclusion, and equality values through physical education. Undoubtedly, this proposal is of great importance as implementing those values is fundamental for the person and society. In general, this study is original and of great importance. The writing, structure, and level of clarity are excellent. References are correctly presented both in-text and in the reference section.

Author Response

Thank you very much for your time and effort in reviewing our work. Reading such compliments to our work makes us really happy! 

In regards to English language and style, we have reviewed the whole document and rewritten some sentences in order to facilitate comprehension of the readers. 

Again, thank you very much for your words and for your effort and time in difficult dates like these, when everyone is busy and lacks of time. 

Best regards, 

The authors

Reviewer 2 Report

Comments to the authors

GENERAL COMMENTS

The study is very interesting. But I think it is insufficient to be published in a JCR Q1 journal. Presenting the intervention program is not enough.

It is necessary to show the results obtained after applying the intervention program in a specific sample.

The intervention program itself does not guarantee success. It is possible to implement the program and not achieve the expected results (the acquisition of values by the students).

Therefore, it is very important to carry out a thorough revision of the manuscript showing the results achieved by applying the intervention program.

Below, I give a series of recommendations that can improve the quality of the manuscript in this regard.

SPECIFIC COMMENTS

  1. Introduction

The authors make a good analysis of the values transmitted by sport and physical activity through physical education classes.

They approach the concept of values from a sociological, psychological and educational perspective.

The authors point out the agents that drive the development of values in people/students: family, friends, school, etc., and therefore in physical education classes.

The authors clearly point out that “The main objective of this work is to create an intervention proposal which aims to promote, following a set of intervention units of sessions, the development of the three main values in which this work is based: respect, equality and inclusion”. Therefore, the authors focus their efforts on clarifying in the introduction these three values (respect, equality and inclusion), and how physical education and the use of the Teaching Personal and Social Responsibility Model (TPSR) can promote their achievement.

Therefore, the authors conceptualize the TPSR and its four key points: core values, levels of responsibility, program leader responsibilities and daily program format. As well as some studies that have used the TPSR.

  1. 2. Materials and Methods

The information provided in lines 210 to 225 should form a new subsection of the method called "Procedure". Since the information indicated therein relates the steps followed to develop the present study.

2.1 Systematic review

I think this section is explained in a very simple way. But the information provided shows that this is a weak point of the work.

I don't quite understand why they only limited the reference search to the SCOPUS, SPORTDISCUS and ERIC databases.

How could it be that they did not perform a Web of Science search?

There are very noticeable absences in your search strategy. I understand the keywords you have used for your search, and how they are related to the three values of your study (respect, equality and inclusion). But I do not understand how you did not use "Teaching Personal and Social Responsibility Model" for your search.

I did a quick search in Web of Science with the keyword "Teaching Personal and Social Responsibility Model" as title and 21 papers appeared. Among these, there is a systematic review of programs based on the teaching personal and social responsibility model-based programs in physical education ("Teaching personal and social responsibility model-based programmes in physical education: A systematic review"). And this reference does not even appear in its list of references.

I think this may have conditioned the quality of their work. They could have developed a more detailed review of the existing literature. Perhaps it would have improved the quality of your discussion by being able to contrast your work with other existing work.

Lines 248-250.

The sample object of this project is not well described. It is necessary to indicate several aspects:

  • It should be noted if the intervention program was applied in one center, several centers (public, private, or concerted).
  • How many students did it apply to (indicate how many men and how many women), as well as the mean age of the sample (Mean ± SD).
  • Experimental death (in how many students the intervention program was not finally applied and why). Indicate the final sample to which the intervention program was applied.
  • Furthermore, taking into account that " TPSR arose in the United States, and its main objective was to achieve a teaching methodology that could transmit values and life skills to socially excluded young people " it would be important to know what percentage of students were at risk of social exclusion
  • Clarify the ethical aspects of the study: informed and signed consents of the participants or their tutors, approval of the ethics committee, etc.
  • Create a sub-section called "Participants" in the method that contains all this information.

Line 306 lacrosse. I would like to know if they used lacrosse or intercrosse. Lacrosse is characterized by extreme physical contact. On the contrary, intercrosse is an adaptation of non-contact lacrosse created for use in physical education classes. If intercrosse was used, it is necessary to substitute in different parts of the text.

Two sub-sections are missing in the method:

  • Measuring instruments
  • Statistical analysis

  1. 3. Results

This section provides a detailed description of the intervention program applied..

Lines 299-302, the authors state the following:

After the brief explanation, each student will be given a questionnaire that he/she will also have to fill in before starting and at the end of the intervention. Specifically, Personal and Social Responsibility Questionnaire (PSRQ) by Li et al [40], in its Spanish version [41], will be used.

I do not understand why they use a questionnaire at the beginning and end of the session if they do not show the results of this questionnaire. I think that this is precisely what should appear in the results section (with tests, retest), differentiating between sexes and by grades (if the sample were large enough).

Lines 373-378, the authors state the following:

In this project, like Pardo [25], a personalized document is used for each participant, which requires dedication and special attention throughout the intervention: the diaries. The diaries will be documents drawn up by the students. For each session carried out, they should describe the ideas, reflections, emotions and thoughts that arose from their experience in the practice of the different sports modalities. In addition, a self-evaluation section will be added, which addresses behavior and participation during the session.

I also do not understand why the results of these diaries have not been presented.

In my humble opinion all the information currently contained in the results section is a description of the intervention program. This information should be in the method section, as a new subsection called "Intervention program". It could even go as supplementary material.

And the "results" should include the results obtained by the students in the measurement instruments used:

  • Specifically, Personal and Social Responsibility Questionnaire (PSRQ), version Española.
  • Student's diary.

These two measurement tools should be explained in the methods section in a new subsection called "Measuring instruments".

  1. 4. Discussion and conclusions

This section does not really include a discussion of the results, but rather different reflections on the intervention program.

This is due to the fact that in the results section there are not the results I have mentioned, but only the intervention program is described.

References

Although there is a high number of references (52), there is a lack of references to the Teaching Personal and Social Responsibility Model (TPSR).

Round 2

Reviewer 2 Report

After reading the authors' responses, it is clear that they have not made an intervention. The authors develop an intervention program.
Therefore, many of my proposals were not possible to solve.
In spite of this, the authors conveniently resolve the rest of the comments or suggestions.
Therefore, I consider that the manuscript should be accepted.